# Incidence and prevalence of inflammatory bowel disease in UK primary care: a population-based cohort study

Thomas Joshua Pasvol ![ORCID],[1] Laura Horsfall,[1] Stuart Bloom,[2] Anthony Walter Segal,[3] Caroline Sabin,[4,5] Nigel Field,[4,6] Greta Rait[1,5]

For numbered affiliations see end of article.

**Correspondence to**
Dr Thomas Joshua Pasvol;
thomas.pasvol@nhs.net

## ABSTRACT

**Objectives** We describe temporal trends in the recorded incidence of inflammatory bowel disease (IBD) in UK primary care patients between 2000 and 2018.

**Design** A cohort study.

**Setting** The IQVIA Medical Research data (IMRD) primary care database.

**Participants** All individuals registered with general practices contributing to IMRD during the period 01 January 2000–31 December 2018.

**Main outcome measures** The primary outcome was the recorded diagnosis of IBD.

**Results** 11 325 025 individuals were included and 65 700 IBD cases were identified, of which 22 560 were incident diagnoses made during the study period. Overall, there were 8077 incident cases of Crohn's disease (CD) and 12 369 incident cases of ulcerative colitis (UC). Crude incidence estimates of 'IBD overall', CD and UC were 28.6 (28.2 to 28.9), 10.2 (10.0 to 10.5) and 15.7 (15.4 to 15.9)/100 000 person years, respectively. No change in IBD incidence was observed for adults aged 17–40 years and children aged 0–9 years. However, for adults aged over 40 years, incidence fell from 37.8 (34.5 to 41.4) to 23.6 (21.3 to 26.0)/100 000 person years (average decrease 2.3% (1.9 to 2.7)/year (p<0.0001)). In adolescents aged 10–16 years, incidence rose from 13.1 (8.4 to 19.5) to 25.4 (19.5 to 32.4)/100 000 person years (average increase 3.0% (1.7 to 4.3)/year (p<0.0001)). Point prevalence estimates on 31 December 2018 for IBD overall, CD and UC were 725, 276 and 397 per 100 000 people, respectively.

**Conclusions** This is one of the largest studies ever undertaken to investigate trends in IBD epidemiology. Although we observed stable or falling incidence of IBD in adults, our results are consistent with some of the highest reported global incidence and prevalence rates for IBD, with a 94% rise in incidence in adolescents. Further investigation is required to understand the aetiological drivers.

## INTRODUCTION

The inflammatory bowel diseases (IBD; Crohn's disease (CD), ulcerative colitis (UC) and IBD unclassified (IBDU)) are chronic inflammatory conditions of unknown aetiology that affect the gastrointestinal tract.[1 2] In North America, over 1.5 million individuals are living with IBD and in Europe it is estimated that 2.5–3 million individuals are affected, with an estimated direct healthcare cost of 4.6–5.6 billion Euros/year.[3 4] Historically, IBD was regarded as a disease of high-income western countries with a substantial rise in incidence observed during the latter half of the 20th century.[5] However, there is evidence that the rate has plateaued in western countries while rising rapidly in newly industrialised countries.[4] Accurate and up-to-date estimates of trends in incidence and prevalence of IBD are an essential step in preparing services for the delivery of future IBD care.

Studies using local hospital records and secondary care databases have been conducted to describe the epidemiology of IBD in the UK.[6–9] However, patient follow-up is challenging and loss to follow-up may introduce bias, notably where patients do

not require hospitalisation and/or move geographical location. More recently, estimates of incidence and prevalence of IBD were reported in a rigorously validated IBD cohort of 10 926 cases in Lothian, Scotland.[10] However, it remains unknown if these findings are generalisable across the UK.

In the UK, over 60 000 000 people are registered with a general practice (GP; about 91% of the population).[11] Electronic GP health records databases can enable large-scale investigation of relatively rare diagnoses such as IBD.[12] The largest such UK study performed to date was undertaken in Northern England and included 179 incident cases of IBD diagnosed in a population of 135 723 during the period 1984–1995.[13]

In the present study, we investigated temporal trends in the incidence of IBD diagnoses from 2000 to the end of 2018 using electronic GP data from the IQVIA Medical Research Data (IMRD) primary care database (formally The Health Improvement Network database).

## MATERIALS AND METHODS
### Data source
IMRD is a large longitudinal database currently containing the anonymised electronic medical records of 18.3 million patients collected from 797 GPs throughout the UK; 3 million of these patients are presently registered with a practice contributing to IMRD and are currently providing data. It is one of the most comprehensive data sources of its kind and is used worldwide for research by academic institutions, government departments and the pharmaceutical industry. Data are based on patient consultation records and include demographics (eg, gender, age and socioeconomic level of deprivation), presenting symptoms and diagnoses, referrals to secondary care, medications, results of investigations, vaccinations and additional health data such as height, blood pressure, weight and smoking status. Data are recorded in IMRD using the Read code hierarchical coding system.[14] No other coding system was used in IMRD for the duration of the observation period. The GPs in IMRD are broadly representative of all primary care practices in the UK in terms of age and sex of patients, practice size, geographical distribution and the prevalence of numerous chronic conditions such as hypertension, diabetes, asthma and epilepsy.[15] A previous validation study using electronic GP records showed that for individuals with a code for IBD, the diagnosis was highly probable or probable for 92% (95% CI 86% to 96%) of patients.[12] Although IMRD data are not linked to secondary care records, diagnoses made in secondary care are captured in IMRD; either through letters and communications to the GP or during patient consultations.

### Study population
All data included in this study were from time periods after the GPs had met acceptable computer usage (ACU) and acceptable mortality reporting (AMR) standards.[16 17]

The date of meeting ACU was the date after which the practice was confirmed to have electronically logged an average of at least one medical record, one additional health record and two prescriptions per person year. The date of meeting AMR standards was the date after which the practice was confirmed to record mortality at a similar rate to that expected for a population with comparable demographics as per the Office for National Statistics.

All individuals of any age contributing data between the 1 January 2000 and the 31 December 2018 were included. The study was a dynamic cohort with individuals entering and exiting at different times. Cohort entry was defined as the latest date of the following: 1 January 2000; the date of registration with the GP plus 9 months to account for prevalent disease being recorded as incident disease when patients register with the practice (this time period was selected using previously published methodology)[18]; or the date the practice met predefined quality indicators for electronic data (AMR and ACU). Cohort exit was defined as the earliest date of the following: first diagnosis of IBD; deregistration with the GP contributing data; death; or 31 December 2018.

### Main outcome definitions
The main outcomes of interest were newly diagnosed CD, UC or any IBD. The any IBD category included specific and general terms for IBD (comprising CD, UC, IBDU and unspecified IBD). Read code lists, adapted from those used in previous literature,[12 13 19] were generated for all three main outcomes using published methodology,[20] then subsequently discussed with a panel of experts including gastroenterologists, epidemiologists, a GP and a statistician (online supplementary appendix Read code lists).

As a quality filter, individuals were only included in the study as cases if they had at least two IBD Read codes recorded on separate dates or at least one IBD Read code plus at least one prescription for a drug commonly used to treat IBD (any aminosalicylate or rectal steroid enema listed in chapter 1.5 of the British National Formulary,[21] azathioprine, mercaptopurine, methotrexate, ciclosporin, infliximab or adalimumab) (online supplementary appendix Read code lists). As 'incident IBD cases' (who have not been prescribed 'IBD drugs') were required to have their diagnosis verified on a subsequent GP visit, we anticipated that this would result in under ascertainment of those diagnosed in the final months of the study period. Thus, temporal trends in incidence are graphically presented for the period 2000–2017 as opposed to 2000–2018. The date at which the first recording of any IBD code or IBD drug prescription was made was classified as the incident date. For individuals who had been given a code for both UC and CD in their lifetime, the most recent code recorded was used as their final diagnosis.

A separate algorithm was developed to explore the validity of the diagnosis of IBD in IMRD. This involved checking whether individuals who had ever been given a

medical Read code for IBD had a record of (1) presentation with symptoms suggestive of IBD (abdominal pain, diarrhoea, bloody stools, weight loss); (2) a prescription for a drug commonly used to treat IBD. 'Incident cases' were not required to meet these criteria to be included in the analysis.

## Covariates

The following covariates were included in the analyses: (1) birth gender, (2) age by Montreal/Paris classification[22 23] (A classification system for IBD whereby the 'A' variable describes 'age at diagnosis', the levels of which are: A1a (0–9 years); A1b (10–17 years); A2 (17–40 years) and A3 (40+ years)), (3) calendar time, (4) Townsend Deprivation Index (a quintile measurement of social deprivation based on post code linked census data)[24] and (5) geographical location of GP. This was included at the level of former Strategic Health Authority (a defined region responsible for the management of health services for that particular area) for England and at the level of country for Scotland, Wales and Northern Ireland.

## Statistical analyses

Crude incidence estimates per 100 000 person years at risk were calculated by dividing the total number of cases by the total number of person years of follow-up then multiplying by 100 000. This was done separately for CD, UC and any IBD with 95% CIs estimated assuming a Poisson distribution. Stratified incidence rates were calculated by sex, age, Townsend Deprivation Index and geographical location. Time period was fitted as both a continuous variable and a categorical variable by calendar year. Mixed multivariable Poisson regression was used to estimate incidence rate ratios (IRRs). Individuals with missing data on Townsend score were included in the analysis using 'missing' as a level to the Townsend variable. GP was included as a random effect to account for any data clustering by practice; the other covariates were included as fixed effects. The Wald test was used to test for significance of categorical variables in the regression model and to test for multiplicative interactions.

Point prevalence was calculated by dividing all cases of IBD (both incident and prevalent) by the total number of individuals contributing data to the cohort on the last day of the study period.

StataCorp. 2017. *Stata Statistical Software: Release 15*. College Station, Texas: StataCorp LLC was used for all analyses.

## Sensitivity analysis

In the sensitivity analysis, we broadened our case definition to include any individual who had a single IBD medical Read code (as opposed to two medical Read codes or one medical Read code plus one relevant prescription).

## Patient and public involvement

We involved representatives from the University College Hospitals NHS Foundation Trust patient with IBD panel in the early stages of protocol design. However, we did not involve patient and public involvement (PPI) representatives in other aspects of study design or analysis due to IMRD licence agreements and the technical computer programming methods that were involved. We intend to involve PPI representatives in writing a plain language summary for dissemination to peers and patient groups.

## RESULTS

11 325 025 individuals (78 985 977 person years of follow-up) were included in the cohort. 5 541 508 (48.9%) were male. 7 944 975 (70.0%) were registered with a GP in England, 1 690 503 (14.9%) Scotland, 1 285 722 (11.4%) Wales and 403 825 (3.6%) Northern Ireland. Mean (SD) age at cohort entry was 34 (22.8) years and median (IQR) follow-up was 5.4 (2.0–11.6) years. We identified 65 700 cases of IBD, including 24 991 cases of CD and 36 705 cases of UC. Among these, 22 560 (8077 for CD and 12 369 for UC) were incident diagnoses made during study follow-up. Overall, crude incidence estimates were 28.6 (95% CI 28.2 to 28.9), 10.2 (95% CI 10.0 to 10.5) and 15.7 (95% CI 15.4 to 15.9)/100 000 person years for 'any IBD', CD and UC, respectively. Point prevalence estimates on 31 December 2018 were 725, 276 and 397 per 100 000 people for 'any IBD', CD and UC, respectively.

Of 28 879, 24 173 (83.7%) individuals given a new code for IBD since entering the study had a record of a prescription for a drug commonly used to treat IBD, comparing to 1.8% for the whole cohort. Additionally, 23 337/28 879 (80.8%) with a code for IBD had a record of presentation to their GP with either diarrhoea, bloody stools, abdominal pain or weight loss, comparing to 33.6% for the whole cohort.

For the period 2000–2017, incidence of 'any IBD' remained relatively stable for those aged 17–40 years (A2 disease) and those aged 0–9 years (A1a disease). However, for those aged over 40 years (A3 disease), crude incidence fell from 37.8 (95% CI 34.5 to 41.4) to 23.6 (21.3–26.0) at an average rate of 2.3% (95% CI 1.9% to 2.7%) per calendar year (p<0.0001) and for those aged 10–16 years (A1b disease), incidence rose from 13.1 (95% CI 8.3 to 19.5) to 25.4 (95% CI 19.5 to 32.4) at an average rate of 3.0% (95% CI 1.7% to 4.3%) per calendar year (p<0.0001; figure 1). When adding an age–time interaction term to the model, we found an interaction for all three main outcomes (p<0.00001, p=0.0046, p<0.00001 for 'any IBD', CD and UC, respectively). Ageband-specific age–time interaction coefficients confirmed increasing incidence in adolescents ages 10–16, decreasing incidence in those aged 40+ years and stable incidence in age groups 0–9 and 17–40 years (online supplementary appendix table 1).

### CD incidence

During the study period, CD incidence fell slightly from 10.7 (95% CI 9.5 to 12.1) to 9.0 (95% CI 8.0 to 10.1)/100 000 person years at an average rate of 1.0% (95% CI 0.6% to 1.5%) per calendar year (p<0.0001; figure 2).

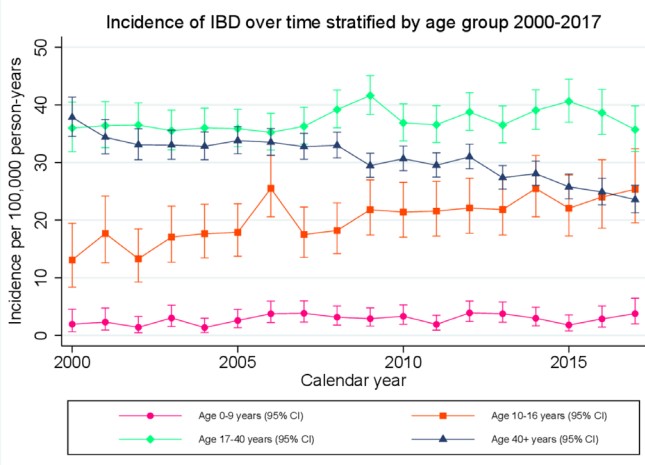

**Figure 1** Crude incidence estimates for any inflammatory bowel diseases (IBD), stratified by Montreal/Paris age classification and calendar year, over the period 2000–2017.

However, in children <17 years, incidence rose from 3.9 (95% CI 2.2 to 6.2) to 6.9 (95% CI 4.9 to 9.3)/100 000 person years at an average rate of 2.9% (95% CI 1.3% to 4.4%) per calendar year (p<0.0001; figure 3). Although overall crude incidence was higher for boys than for girls (7.4 (95% CI 6.8 to 8.0) vs 4.1 (95% CI 3.6 to 4.6)), a significant rise in incidence was observed for both sexes (average 2.7% (95% CI 0.8% to 4.6%) and 3.3% (95% CI −0.6% to 6.0%) rise per calendar year for boys and girls respectively). No change in incidence was observed for children aged 0–9 years. However, for adolescents aged 10–16 years, incidence rose from 7.6 (95% CI 4.2 to 12.8) to 13.1 (95% CI 9.0 to 18.4)/100 000 person years at an average rate of 2.8% (1.2–4.5) per calendar year (p=0.001).

Incidence of CD was highest in Northern Ireland, Scotland and the North West (13.1 (95% CI 12.0 to 14.4), 12.3 (95% CI 11.7 to 12.9) and 11.9 (95% CI 11.1 to 12.7)/100 000 person years, respectively) and lowest in

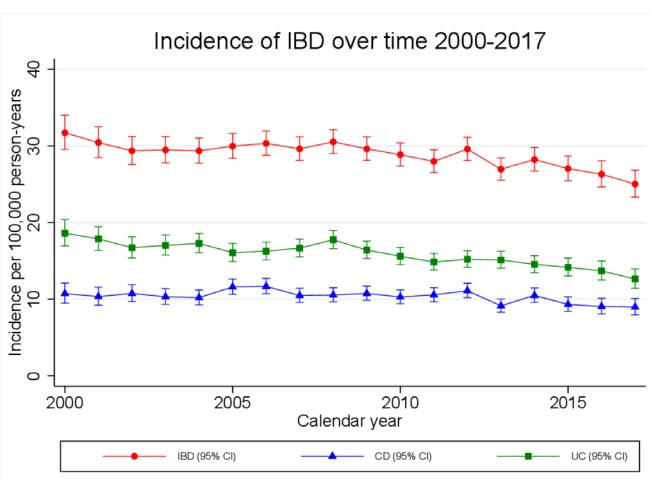

**Figure 2** Crude incidence estimates of inflammatory bowel diseases (IBD), stratified by calendar year, over the period 2000–2017. CD, Crohn's disease; UC, ulcerative colitis.

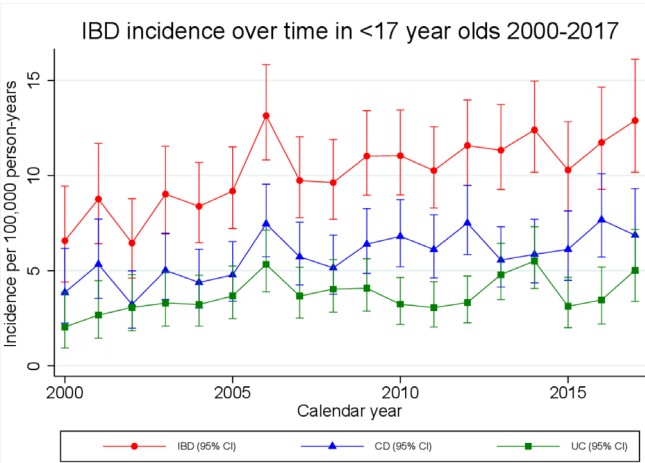

**Figure 3** Crude incidence estimates of inflammatory bowel diseases (IBD) in children <17 years, stratified by calendar year, over the period 2000–2017. CD, Crohn's disease; UC, ulcerative colitis.

Wales, London and the West Midlands (8.9 (95% CI 8.4 to 9.6), 8.9 (95% CI 8.3 to 9.6) and 8.4 (95% CI 7.7 to 9.1)/100 000 person years, respectively, figure 4). We observed no association between social deprivation and incidence of CD after adjusting for sex, calendar year, age and geographical location (table 1).

### UC incidence
Incidence of UC dropped to a greater extent than for CD over the study period; from 18.6 (95% CI 17.0 to 20.4) to 12.6 (95% CI 11.4 to 13.9)/100 000 person years at an average rate of 1.6% (95% CI 1.3% to 2.0%) per calendar year (p<0.0001; figure 2). The fall in incidence was most pronounced for those aged over 40 years, in whom a 45% drop in incidence was observed, falling from 24.1 (95% CI 21.5 to 26.9) to 13.3 (95% CI 11.6 to 15.2)/100 000 person years (average 3.1% (95% CI 2.6% to 3.6%) decrease per calendar year (p<0.0001)).

In children aged <17, incidence rose from 2.0 (95% CI 0.9 to 3.9) to 5.0 (95% CI 3.4 to 7.2)/100 000 person years (average 2.5% (95% CI 0.5% to 4.4%) rise per calendar year (p=0.01); figure 3). The rise in incidence was largely driven by adolescent boys aged 10–16 in whom incidence rose by 3.4% (95% CI 0.8% to 6.2%) per calendar year (p=0.01). No significant change in incidence was observed in girls aged 10–16 years or children of either sex aged 0–9 years.

Incidence of UC was highest in the North East, the East of England and the East Midlands (18.1 (95% CI 16.0 to 20.5), 17.4 (95% CI 16.2 to 18.6) and 17.5 (95% CI 15.6 to 19.6)/100 000 person years, respectively) and lowest in Wales, the South West and London (14.1 (95% CI 13.4 to 14.9), 15.2 (95% CI 14.2 to 16.2) and 15.3 (95% CI 14.5 to 16.2)/100 000 person years, respectively; figure 4). We observed higher incidence of UC in individuals from the least deprived quintile compared with most deprived (16.8 (95% CI 16.2 to 17.5) vs 13.3 (95% CI 12.6 to

**Table 1** Incidence rates and adjusted incidence rate ratios (IRRs) of Crohn's disease (CD) and ulcerative colitis (UC) by sex, age, calendar year, geographical location and social deprivation

| | Incidence of CD | | Incidence of UC | |
| --- | --- | --- | --- | --- |
| | Rate per 100 000 | | Rate per 100 000 | |
| | Person years (95% CI) | Adjusted IRR (95% CI)* | Person years (95% CI) | Adjusted IRR (95% CI)* |
| Overall | 10.2 (10.0 to 10.5) | | 15.7 (15.4 to 15.9) | |
| Sex | | | | |
| Male | 9.3 (0.00 to 9.6) | 1 | 16.7 (16.3 to 17.1) | 1 |
| Female | 11.1 (10.8 to 11.5) | 1.20 (1.15 to 1.25) | 14.7 (14.3 to 15.0) | 0.87 (0.84 to 0.90) |
| Age, years | | | | |
| 0–9 | 1.6 (1.3 to 1.9) | 0.11 (0.09 to 0.13) | 1.0 (0.8 to 1.3) | 0.05 (0.04 to 0.07) |
| 10–16 | 11.5 (10.7 to 12.4) | 0.77 (0.71 to 0.84) | 7.4 (6.8 to 8.1) | 0.38 (0.35 to 0.42) |
| 17–40 | 14.9 (14.4 to 15.4) | 1 | 19.2 (18.7 to 19.8) | 1 |
| 40+ | 9.2 (8.9 to 9.5) | 0.62 (0.59 to 0.64) | 17.9 (17.5 to 18.3) | 0.92 (0.89 to 0.96) |
| Year (linear change) | | 0.99 (0.98 to 0.99) | | 0.98 (0.98 to 0.99) |
| Year (categorical variable) | | | | |
| 2000 | 10.7 (9.5 to 12.1) | 1 | 18.6 (17.0 to 20.4) | 1 |
| 2001 | 10.3 (9.2 to 11.6) | 0.97 (0.82 to 1.14) | 17.9 (16.4 to 19.5) | 0.96 (0.85 to 1.09) |
| 2002 | 10.8 (9.7 to 11.9) | 1.00 (0.85 to 1.17) | 16.7 (15.4 to 18.1) | 0.89 (0.79 to 1.01) |
| 2003 | 10.3 (9.3 to 11.4) | 0.95 (0.82 to 1.11) | 17.0 (15.8 to 18.4) | 0.91 (0.81 to 1.03) |
| 2004 | 10.2 (9.3 to 11.2) | 0.94 (0.80 to 1.09) | 17.3 (16.1 to 18.6) | 0.93 (0.82 to 1.04) |
| 2005 | 11.6 (10.6 to 12.6) | 1.07 (0.92 to 1.24) | 16.1 (14.9 to 17.3) | 0.86 (0.77 to 0.97) |
| 2006 | 11.7 (10.7 to 12.7) | 1.07 (0.93 to 1.24) | 16.3 (15.1 to 17.5) | 0.87 (0.78 to 0.98) |
| 2007 | 10.5 (9.6 to 11.4) | 0.96 (0.83 to 1.12) | 16.7 (15.5 to 17.9) | 0.89 (0.80 to 1.00) |
| 2008 | 10.6 (9.7 to 11.5) | 0.97 (0.84 to 1.13) | 17.8 (16.6 to 19.0) | 0.95 (0.85 to 1.07) |
| 2009 | 10.8 (9.9 to 11.7) | 0.99 (0.86 to 1.15) | 16.4 (15.3 to 17.6) | 0.88 (0.79 to 0.99) |
| 2010 | 10.3 (9.4 to 11.2) | 0.95 (0.82 to 1.10) | 15.6 (14.5 to 16.8) | 0.84 (0.75 to 0.94) |
| 2011 | 10.6 (9.7 to 11.5) | 0.97 (0.84 to 1.13) | 14.9 (13.8 to 16.0) | 0.80 (0.71 to 0.90) |
| 2012 | 11.1 (10.2 to 12.1) | 1.02 (0.88 to 1.18) | 15.2 (14.2 to 16.3) | 0.82 (0.73 to 0.92) |
| 2013 | 9.1 (8.3 to 10.0) | 0.84 (0.72 to 0.97) | 15.1 (14.0 to 16.2) | 0.82 (0.73 to 0.92) |
| 2014 | 10.5 (9.6 to 11.5) | 0.96 (0.82 to 1.11) | 14.5 (13.5 to 15.7) | 0.79 (0.70 to 0.89) |
| 2015 | 9.3 (8.4 to 10.3) | 0.84 (0.72 to 0.98) | 14.2 (13.0 to 15.4) | 0.77 (0.68 to 0.87) |
| 2016 | 9.0 (8.1 to 10.1) | 0.80 (0.68 to 0.95) | 13.7 (12.5 to 15.0) | 0.74 (0.65 to 0.84) |
| 2017 | 9.0 (8.0 to 10.1) | 0.79 (0.67 to 0.94) | 12.6 (11.4 to 13.9) | 0.68 (0.60 to 0.78) |
| 2018 | 5.9 (5.1 to 6.9) | 0.52 (0.43 to 0.63) | 10.0 (8.9 to 11.2) | 0.54 (0.47 to 0.63) |
| Region | | | | |
| East Midlands | 9.6 (8.2 to 11.2) | 1 | 17.5 (15.6 to 19.6) | 1 |
| East of England | 10.9 (10.0 to 11.9) | 1.18 (0.96 to 1.44) | 17.4 (16.2 to 18.6) | 1.01 (0.86 to 1.19) |
| London | 8.9 (8.3 to 9.6) | 0.94 (0.77 to 1.14) | 15.3 (14.5 to 16.2) | 0.94 (0.81 to 1.09) |
| North East | 9.8 (8.2 to 11.6) | 1.03 (0.80 to 1.34) | 18.1 (16.0 to 20.5) | 1.08 (0.88 to 1.32) |
| North West | 11.9 (11.1 to 12.7) | 1.27 (1.05 to 1.54) | 15.8 (14.9 to 16.7) | 0.93 (0.80 to 1.08) |
| Northern Ireland | 13.1 (12.0 to 14.4) | 1.43 (1.16 to 1.75) | 16.2 (14.9 to 17.5) | 1.02 (0.87 to 1.21) |
| Scotland | 12.3 (11.7 to 12.9) | 1.36 (1.13 to 1.64) | 15.3 (14.7 to 16.0) | 0.97 (0.84 to 1.13) |
| South Central | 9.5 (8.9 to 10.2) | 1.01 (0.83 to 1.23) | 16.1 (15.2 to 17.0) | 0.92 (0.79 to 1.07) |
| South East Coast | 9.4 (8.7 to 10.1) | 1.04 (0.85 to 1.26) | 15.7 (14.9 to 16.6) | 0.95 (0.81 to 1.10) |
| South West | 10.1 (9.3 to 11.0) | 1.08 (0.89 to 1.32) | 15.2 (14.2 to 16.2) | 0.88 (0.75 to 1.03) |

**Table 1** Continued

| | Incidence of CD | | Incidence of UC | |
|---|---|---|---|---|
| | **Rate per 100 000** | | **Rate per 100 000** | |
| | **Person years (95% CI)** | **Adjusted IRR (95% CI)\*** | **Person years (95% CI)** | **Adjusted IRR (95% CI)\*** |
| Wales | 8.9 (8.4 to 9.6) | 0.98 (0.81 to 1.19) | 14.1 (13.4 to 14.9) | 0.87 (0.75 to 1.01) |
| West Midlands | 8.4 (7.7 to 9.1) | 0.89 (0.73 to 1.09) | 16.2 (15.3 to 17.2) | 0.96 (0.82 to 1.12) |
| Yorkshire and Humber | 9.6 (8.2 to 11.2) | 1.00 (0.78 to 1.28) | 15.8 (14.0 to 17.8) | 0.91 (0.74 to 1.10) |
| Townsend, quintile | | | | |
| Missing | 10.1 (9.6 to 10.7) | 1.08 (0.99 to 1.17) | 15.1 (14.4 to 15.7) | 0.95 (0.89 to 1.02) |
| 1 | 9.6 (9.1 to 10.1) | 1 | 16.8 (16.2 to 17.5) | 1 |
| 2 | 10.0 (9.5 to 10.5) | 1.02 (0.95 to 1.10) | 16.5 (15.9 to 17.2) | 0.97 (0.92 to 1.03) |
| 3 | 10.7 (10.1 to 11.2) | 1.07 (1.00 to 1.15) | 16.2 (15.5 to 16.8) | 0.96 (0.91 to 1.01) |
| 4 | 10.4 (9.8 to 11.0) | 1.03 (0.96 to 1.12) | 14.8 (14.1 to 15.5) | 0.88 (0.83 to 0.94) |
| 5 | 11.2 (10.5 to 11.9) | 1.08 (0.99 to 1.17) | 13.3 (12.6 to 14.1) | 0.80 (0.74 to 0.86) |

\*Adjusted for other variables considered: sex, ageband, year, region, Townsend quintile, respectively; IRRs compared with the reference group for each categorical variable.

14.1)/100 000 person years, adjusted IRR 0.80 (95% CI 0.74 to 0.86), table 1).

### Sensitivity analysis

When broadening the case definition to include any individual who had a single IBD medical Read code, we observed overall incidence rates of 36.6 (95% CI 36.2 to 37.0), 12.9 (95% CI 12.7 to 13.2) and 19.3 (95% CI 19.0 to 19.6)/100 000 person years for 'any IBD', CD and UC, respectively. We observed a similar fall in incidence of UC, decreasing from 21.8 (95% CI 20.0 to 23.8) to 17.9 (95% 16.4 to 19.4)/100 000 person years (average decrease 1.3% (95% CI 0.9% to 1.6%) per calendar year). However, no fall in CD incidence was observed (online supplementary appendix figure 1). When stratifying IBD incidence by 5 year age bands, the peak in incidence later in life was higher and occurred later than in the primary analysis (online supplementary appendix figures 2 and 3).

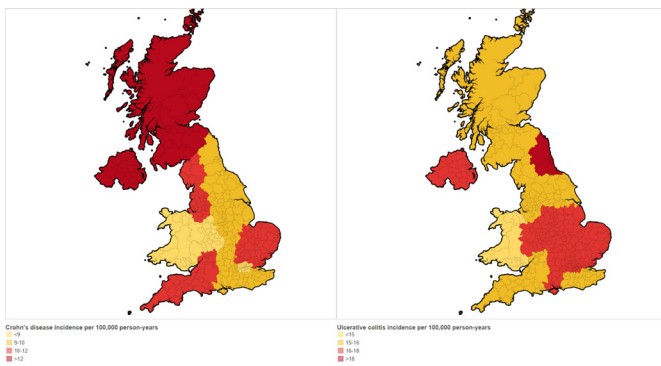

**Figure 4** Map showing overall crude incidence of Crohn's disease and ulcerative colitis stratified by geographical region.

### DISCUSSION

This is one of the largest observational studies undertaken to investigate trends in IBD epidemiology. Although incidence of IBD remained relatively stable for those aged 17–40 years and those aged 0–9 years, we observed a 38% fall in incidence for those aged over 40 years and a 94% rise in incidence in the adolescent population. The most recent incidence estimates are in line with some of the highest reported rates of paediatric IBD internationally.[25–27]

Study strengths include the large sample size and the prospective collection of healthcare records representative of 'real-life' clinical practice. Unlike previous incidence/prevalence studies that have relied on external data sources to estimate denominator population characteristics, we were able to extract demographics and person-time follow-up for all individuals in our cohort, including those who did not develop IBD. Additionally, IMRD has been shown to be broadly representative of the UK in terms of age, sex, mortality rates and prevalence of numerous comorbidities,[15] allowing us to draw inferences from our data and relate this to the UK population as a whole. Not only has the diagnosis of IBD been validated in a similar GP database,[12] but we have demonstrated that the majority of individuals coded for IBD in IMRD have been prescribed drugs commonly used to treat IBD and presented with symptoms in keeping with IBD. This would support the argument that IMRD represents an important and useful resource for further epidemiological studies of IBD.

Limitations arise when conducting GP database research, particularly as the primary use of the software that contributes to IMRD is for patient management purposes rather than medical research. Thus, data can be incomplete and will often only reflect those events

that are deemed to be relevant to the patient's care. Given that we were also reasonably strict with our case definition, this may have resulted in underascertainment of cases. Although we find reason to be confident in the validity of the data, we were not able to confirm our cases by evidence of radiological, endoscopic or histological findings. Therefore, it is possible that some individuals were misclassified. There was a small risk of duplication of medical records. This could occur if a patient deregistered with one practice contributing to IMRD then subsequently registered with another IMRD practice during the observation period. This is likely to be the case for a very small number of individuals as IMRD only covered 5%–6% of UK GP during the study period. Although the total number of individuals contributing may be a slight overestimate, this would have no effect on incidence or prevalence rates. This is for two reasons: (1) duplicated records would cover different time periods during the study without overlap; (2) we took steps to ensure that prevalent cases of IBD newly transferring to practices were not counted as incident cases[18] Therefore, incident cases were not counted twice.

Data from a multicentre European study (including two UK sites in North West London and East Yorkshire) reported site incidence rates of 2.6 and 8.4/100 000 person years for CD and 15.9 and 8.2/100 000 years for UC.[28] However, only a small number of UK cases were included (n=167). Incidence rates of 8.3 (3.4 to 13.2) and 13.9 (95% CI 7.5 to 20.3)/100 000 person years for CD and UC, respectively, have been reported in North-East England for the period 1990–1994.[13] We report overall incidence rates of 10.2 (95% CI 10.0 to 10.5) and 15.6 (95% CI 15.3 to 15.9)/100 000 person years for CD and UC, respectively, in a far larger cohort and at a national level. We report considerable geographical variation in IBD incidence across the UK with notably high CD incidence in Scotland and Northern Ireland and high UC incidence in the East of England. This may reflect variation in lifestyle factors such as dietary habits and importantly smoking (it is estimated that 14.4% adults in England smoke compared with 15.9% in Scotland and 16.3% in Northern Ireland).[29]

A Danish study based on nationwide registry data (1995–2012) observed comparable incidence rates: 8.9 (95% CI 8.3 to 9.5) and 10.3 (95% CI 9.7 to 11.0)/100 000 person years for CD and 23.4 (95% CI 22.4 to 24.5) and 23.2 (95% CI 22.2 to 24.3)/100 000 person years for UC in males and females, respectively.[30] In contrast to our results, they observed overall rising incidence rates of IBD, but their study was conducted in a different country over an earlier time period including the 1990s when a rise in IBD incidence was described in many high-income countries. Although they adjusted for age in their analysis, temporal trends in incidence stratified by age group were not reported.

In the sensitivity analysis, we observed higher than expected overall incidence rates. Additionally, for UC, the observed peak in incidence for older individuals was higher than the peak in incidence for younger individuals (online supplementary appendix figure 2); this would be unusual in clinical practice. An explanation for this could be that a number of these patients, who perhaps had colitis of a different aetiology, had been misclassified as IBD. On the basis of these findings, one Read code alone was deemed not specific enough for the diagnosis of IBD.

In keeping with published literature, we observed a rising incidence of paediatric IBD during the early 21st century[31] and we provide further evidence of male preponderance in paediatric IBD when compared with adult onset disease.[32 33] Uniquely, in our study, we have demonstrated rising incidence of adolescent IBD in the context of stable incidence in those aged 17–40 years and falling incidence in the over 40s. This may represent a general shift towards earlier diagnosis of IBD for all age groups except the very young (age 0–9 years). Given that IBD most commonly presents in the second to fourth decade of life,[1 2] rising incidence in adolescents might be explained by a number of factors, including improvements in referral pathways and the introduction of new diagnostic tools (eg, faecal calprotectin testing or capsule endoscopy) resulting in cases being picked up earlier. However, one would expect stepwise increases in incidence when new diagnostic tools are rolled out, which we did not observe. Moreover, if rising incidence of adolescent IBD is due to improved referral pathways, a corresponding rise in incidence might be expected in very young children as well as adolescents. It could be argued that changes in GP coding practice may be contributing. But again, one would expect comparable changes in younger age groups if this were the case. On the other hand, if the epidemiological patterns we observed reflect real increases in the incidence of pathology, this is of great concern and could represent earlier manifestation of disease related to environmental exposures in childhood and adolescence.

Our prevalence estimates were very similar to those reported in a well-validated IBD cohort in Lothian, Scotland[10]; our estimate of IBD prevalence for Scotland on 31 August 2018 was 810 per 100 000 compared with 832 per 100 000 reported by their group. Although our study lacked linkage to secondary care records, the similar prevalence estimates would support the argument that few cases were missed. In 2018, 67 150 000 people were estimated to be living in the UK from which we might extrapolate from our data that there were approximately 487 000 people living with IBD in the UK at that time.

Compounding prevalence of IBD has been demonstrated in Canada and in Scotland.[10 34] This relates to the principle that although IBD incidence may be static or falling, while IBD mortality remains very low, overall prevalence will increase (more people are being diagnosed than are dying). In Scotland, IBD prevalence is estimated to reach 1.0% by 2028. We report rising incidence rates of IBD in younger populations and falling incidence in older age groups. Thus, not only will services need to be attuned to rising IBD prevalence and an ageing

demographic, but also to increasing numbers of new diagnoses in young people who will require lifelong care. This is in the context of significant financial challenges to health services.

## CONCLUSION

Although we observed a stable or falling incidence of IBD in adults over an 18-year period, our results are consistent with some of the highest reported global incidence and prevalence rates for IBD, with a 94% rise in incidence in adolescents. These findings are concerning and suggest that detailed prospective studies are required to understand the aetiological drivers.

**Author affiliations**
¹The Research Department of Primary Care and Population Health, University College London, London, UK
²Medicine, University College London Hospitals NHS Foundation Trust, London, UK
³Medicine, University College London, London, UK
⁴Institute for Global Health, University College London, London, UK
⁵Health Protection Research Unit in Blood Borne and Sexually Transmitted Infections, NIHR, London, UK
⁶Centre for Molecular Epidemiology and Translational Research, University College London, London, UK

**Contributors** TJP: study concept and design; analysis and interpretation of data; drafting of the manuscript; statistical analysis; obtained funding. LH: study concept and design; analysis and interpretation of data; critical revision of the manuscript for important intellectual content; statistical analysis. SB and NF: study concept and design; critical revision of the manuscript for important intellectual content. AWS: study concept and design; critical revision of the manuscript for important intellectual content; obtained funding. CS: study concept and design; critical revision of the manuscript for important intellectual content; statistical analysis. GR: study concept and design; analysis and interpretation of data; critical revision of the manuscript for important intellectual content; study supervision. The corresponding author attests that all listed authors meet authorship criteria and that no others meeting the criteria have been omitted.

**Funding** This work was supported by The Charles Wolfson Charitable Trust (grant number 539234) and The Harbour Foundation (grant number 549321). Laura Horsfall is supported by a Wellcome Trust Fellowship (grant number 209207/Z/17/Z).

**Map disclaimer** The depiction of boundaries on this map does not imply the expression of any opinion whatsoever on the part of BMJ (or any member of its group) concerning the legal status of any country, territory, jurisdiction or area or of its authorities. This map is provided without any warranty of any kind, either express or implied.

**Competing interests** TJP and AWS report research grants from The Charles Wolfson Charitable Trust and The Harbour Foundation for the submitted work.

**Patient and public involvement** Patients and/or the public were involved in the design, or conduct, or reporting, or dissemination plans of this research. Refer to the Methods section for further details.

**Patient consent for publication** Not required.

**Ethics approval** IMRD data collection was approved by the NHS South-East Multicentre Research Ethics Committee in 2003. This study was approved by the Scientific Research Committee (SRC) on 29/09/2018 (SRC reference 18THIN082).

**Provenance and peer review** Not commissioned; externally peer reviewed.

**Data availability statement** Due to licence agreements with IQVIA, we are unable to share patient level data from the IQVIA Medical Research Database. However, we are happy to share our data extractions upon reasonable request. Data requesters should email the corresponding author to request the relevant data.

**ORCID iD**
Thomas Joshua Pasvol http://orcid.org/0000-0002-8334-9931

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
