## [Reviewer comments · BMJ Open]

ARTICLE DETAILS

TITLE (PROVISIONAL)	Incidence and prevalence of inflammatory bowel disease in UK primary care: a population based cohort study
AUTHORS	Pasvol, Thomas; Horsfall, Laura; Bloom, Stuart; Segal, Anthony; Sabin, Caroline; Field, Nigel; Rait, Greta

VERSION 1 - REVIEW

REVIEWER	Takashi Ishige Gunma University Graduate School of Medicine, Japan
REVIEW RETURNED	05-Feb-2020

GENERAL COMMENTS	The authors describes temporal trend of incidence and prevalence of IBD in UK, using administrative health data. This data contains huge number of patients' data, and that seems enough to represent accurate incidence of IBD. The authors suggest decreasing incidence of IBD over the observation period, except for those with pediatric onset. These data are important for future estimates of IBD epidemiology, and the reviewer recommends the manuscript to be accepted. Here are some suggestions; 1. When using administrative data, registration rate and diagnosing rate might change when the operation rules or systems are renewed. The authors should mention that the IMRD data and its coding has not been changed during the observation period. Also, the author should try to show that the registration rate for GP (Page4, introduction: line15) has not been changed over the observation period.2. Discrepancy in trend of incidence among age groups is highly informative in this study. Adding a graph which shows trend of the incidence/prevalence by age groups are recommended.3. Are there change in male-to-female ratio over time? Mionr suggestions: 1. Page 6, Statistical analyses: Please add name and the address of the company produced the statistical software used in this study.2. Male predominance of pediatric IBD is shown in previous studies. Please try to add references.3. Temporal trend of IBD incidence in pediatric population is systematically reviewed by Benchimol E et al [PMID: 20564651]. Adding to the references is recommended.
---

REVIEWER	Jennie Clough NIHR Biomedical Research Centre, Guy's Hospital, London, UK
REVIEW RETURNED	22-Mar-2020

GENERAL COMMENTS	This is a well-written and clear study which benefits from a large sample size. It would benefit from clarification on a couple of points and from minor amendments.  1. Could the authors clarify whether there was any risk of duplication of medical records, and whether steps were taken to reduce duplicates? 2. Although the absolute number of GP practices that use IQVIA is given, what proportion of practices does this represent? 3. The authors indicate that of the 18.3 million patients included on IMRD three million are presently registered and providing data. What is the status of the other 15.3 million patients? 4. Table 1 - please give the full definition of 'IRR' 5. Figure 1 and Figure 3 - can the authors explain the apparent peak in incidence in 2006? 6. Can the authors offer some suggestion as to the varying geographical incidence? Why might incidence be rising more quickly in different regions for UC vs. CD? Or in the devolved nations vs. rest of the UK for CD?
---

REVIEWER	Sarah Nevitt University of Liverpool, United Kingdom
REVIEW RETURNED	01-Apr-2020

GENERAL COMMENTS	I have conducted a statistical review of the manuscript "Incidence and prevalence of inflammatory bowel disease in UK primary care: a population based cohort study" The authors describe a large cohort study investigating trends over time and across age groups in the epidemiology of inflammatory bowel disease including data from over 11 million individuals from GP practices contributing to the IQVIA Medical Research Data primary care database. I consider the statistical analysis approach to be appropriate and well-conducted for the study design and the objectives. Results are mostly well presented (I particularly like the data visualisation on Figure 4) and the discussion is well written, clearly highlighting limitations of the analyses. I have a few comments on the presentation of the results, and around some of the wording in the methods which may require some clarification 1) Page 4, line 24 to 27: "The date of meeting ACU was the date after which the practice was confirmed to have an average of at least one medical record, one additional health record and two prescriptions per person-year." I think something is missing from this sentence relating to acceptable computer usage – presumably this should relate to
---

	medical records, health records and/or prescriptions being computerised? 2) Page 4, line 31: “The study was a dynamic cohort with individuals entering and exiting at different times” and line 36-37: “Cohort exit was defined as the earliest date of the following: first diagnosis of IBD; de-registration with the GP practice contributing data....” Could an individual enter and exit the cohort multiple times (e.g. with unique individuals identified by a single anonymised ID?) for example if an individual left one GP practice and registered at another? 3) Outcome definitions according to the algorithm – presumably this algorithm did not diagnose IBD based on the prescriptions alone (i.e. point 2) and a Read code of symptoms was required (i.e. point 1)? The paragraph below referring to a ‘quality filter’ suggests that first the algorithm validated the diagnosis (based on points 1 or 2 or both?) and then this quality filter checks what the algorithm has diagnosed? I don’t think this is the approach taken, I think the definition here is actually what the ‘quality filter’ apparently does: i.e. “individuals were only included in the study as cases if they had at least two IBD Read codes recorded on separate dates or at least one IBD Read code plus at least one prescription for a drug commonly used to treat IBD.” So to summarise, the descriptions of the algorithm and the quality filter are somewhat confusing. Could these paragraphs be clarified please? 4) Statistical analysis, page 5, line 16-17: “Time period was fitted as both a continuous variable and a categorical variable by calendar year. Mixed multivariable Poisson regression was used to estimate incidence rate ratios (IRRs).” Please clarify how time period was analysed as a continuous variable, I cannot see any results for this within the results section? Within the mixed Poisson regression model, if GP practice was included as a random effect, were other covariates included as fixed-effects? 5) Results: Table 1 and Figures 1 to 3 present incidence rates and IRRs of age groups and by calendar year yet the main findings of this study seem to relate to changes in incident rates over time and within age groups. Some numerical results are quotes for these findings in the text only and the key finding mentioned within the discussion and conclusion seems to be “we observed a 38% fall in incidence for those aged over 40 years and a 94% rise in incidence in the adolescent population” – yet I can’t find the numerical results which relate to this finding in the results section? I would like to see more detail of the age-time interactions described within the text added to Table 1 (or an additional table if there is not enough space), also including the numerical values of these interaction analyses for the age groups where the incident rates remain ‘stable’ for comparison and for completeness. 6) Results: It may be helpful to include a flow diagram or similar to
--	---

	help with visualising the numbers at the start of the results relating to the total number in the cohort, the number of cases and the number of incident cases (of any IBD, of CD and of UC) I also don't understand the difference between the 22,560 incident diagnoses mentioned in the first paragraph of the results and the 28,879 individuals given a new code for IBD. Please clarify the difference between these numbers and these definitions. 7) Table 1: The following additions would be helpful for interpretation: The number of cases with within the covariate subgroups (i.e. how many males and females? How many people in each age group etc.) A footnote to add the interpretation of the adjusted IRR – e.g. what does an IRR<1 or >1 correspond to? Also, is there any particular reason that the age group 17-40 is the reference group? Is this part of the Montreal/Paris definition? 8) UC incidence, page 8: Fig.1 is referred to in the first paragraph but I think this should be a reference to Figure 2?
--	---

REVIEWER	FORT Emmanuel University of Lyon 1, France
REVIEW RETURNED	02-Apr-2020

GENERAL COMMENTS	Chi-trend tests could have been used to analyses the trend of the incidence rates by year.
--

REVIEWER	Kristin Sainani Stanford University, U.S.
REVIEW RETURNED	15-Apr-2020

GENERAL COMMENTS	This a large longitudinal study that provides important epidemiologic information about IBD temporal trends in the UK. Strengths include that the dataset covers 91% of the UK population and also includes demographic information. The paper is concise, the figures are helpful, and the statistical analyses look appropriate. Specific comments: 1. Was the algorithm used to identify IBD patients validated? How sensitive and specific is the algorithm for identifying true IBD cases? The authors mention that "the diagnosis of IBD was validated in a similar GP database" but no details are given. It would be helpful to know the possible extent of misclassification. 2. I'm unclear as to how newly diagnosed cases were differentiated from existing cases. A person could have two IBD Read codes or an IBD Read code plus prescription but also have been diagnosed at a previous date (e.g., prior to 2000). Please clarify how cases were determined to be incident cases. 3. Please clarify for readers who are less familiar with IBD: At what age does the disease typically develop? Is it more likely to develop during certain times in the lifespan? If so, could this explain some of the trends? For example, if most people develop the disease when
---

	they are young, perhaps the decreasing incidence rates in 40+ and increasing rates in 10-16 year-olds simply reflects improved diagnosis whereby true cases are being caught earlier. Please provide this background information for readers who are less familiar with IBD. 4. I found these two sentences confusing, so would suggest rewriting these for clarity: "The phenomenon of compounding prevalence of IBD has been demonstrated in Canada and in Scotland.[10, 32]. This relates to the principle that although IBD incidence may be static or falling, whilst IBD mortality remains very low, prevalence will increase; In Scotland IBD prevalence is estimated to reach 1.0% by 2028."
--	--

REVIEWER	Yue-Fang Chang University of Pittsburgh, USA
REVIEW RETURNED	15-Apr-2020

GENERAL COMMENTS	Comments:  • Please provide a sample description of the 11,325,025 individuals involved in the analyses (e.g. numbers of incident cases, prevalent cases and no diagnosis; numbers of males and females; age at study entry, age at diagnosis, etc.). • Please include the average (and median) follow up time among the prevalent IBD cases, incident IBD cases and those without the diagnosis in the result section. • To benefit the readers, please indicate how the crude incidence rates were estimated in the method section. Were follow up time of those prevalent cases removed from the denominator when calculating the incidence rates? • What was the denominator for the point prevalence calculation (i.e. no. of individuals contributing data to the cohort on the last day of the study period)? Were deceased cases of IBD excluded from the point prevalence calculation? • Were the average rates of increase/decrease per calendar year presented throughout the result section controlled for other covariates listed in the method section? If so, please make it clear.
--

VERSION 1 – AUTHOR RESPONSE

Reviewer: 1

1. When using administrative data, registration rate and diagnosing rate might change when the operation rules or systems are renewed. The authors should mention that the IMRD data and its coding has not been changed during the observation period. Also, the author should try to show that the registration rate for GP (Page4, introduction: line15) has not been changed over the observation period.

Response:

The reviewer makes a valid point that incidence rates may be influenced by changes in coding systems. There were no changes in coding systems during the observation period and this has been included in the revision [lines 74-75]. GP registration rates (and thus the size of our cohort) have changed over the follow up period (in keeping with changes in the background UK population size) and this is taken into account in our incidence calculations (cases per 100,000 person-years at risk).

2. Discrepancy in trend of incidence among age groups is highly informative in this study. Adding a graph which shows trend of the incidence/prevalence by age groups are recommended.

Response:

We agree, please see fig.1

3. Are there change in male-to-female ratio over time?

Response:

The male: female ratio did not change to any great extent over time [please see attached graph]. Given the figure limit of four, we have not included this graph in the manuscript. We didn't identify any important sex interactions.

Minor suggestions:

1. Page 6, Statistical analyses: Please add name and the address of the company produced the statistical software used in this study.
2. Male predominance of pediatric IBD is shown in previous studies. Please try to add references.
3. Temporal trend of IBD incidence in pediatric population is systematically reviewed by Benchimol E et al [PMID: 20564651]. Adding to the references is recommended.

Response:

We agree with all minor suggestions and have amended accordingly [lines 150, 289, 288]

Reviewer 2:

1. Could the authors clarify whether there was any risk of duplication of medical records, and whether steps were taken to reduce duplicates?

Response:

There was a small risk of duplication of medical records. This could occur if a patient de-registered with one practice contributing to IMRD then subsequently registered with another practice contributing to IMRD during the observation period. This is likely to be the case for a small number of individuals as IMRD only covered 5-6% of UK GP practices during the study period. Although the total number of individuals contributing may be a slight overestimate, this would have no effect on incidence or prevalence rates. This is for two reasons; 1) Duplicated records would cover different time periods during the study without overlap 2) We took steps to ensure that prevalent cases of IBD newly transferring to practices were not counted as incident cases [1]. Therefore, an incident case would not be counted twice. [lines 257-263]

2. Although the absolute number of GP practices that use IQVIA is given, what proportion of practices does this represent?

Response:

797 general practices have contributed to IMRD. However, many of these practices no longer exist or have stopped contributing to IMRD. At the end of the study period (31/12/2018) 369/6,993 (5.3%) UK GP practices were contributing to IMRD (<https://www.gponline.com/number-gp-practices-england-falls-below-7000/article/1525443>)

3. The authors indicate that of the 18.3 million patients included on IMRD three million are presently registered and providing data. What is the status of the other 15.3 million patients?

Response:

The remaining patients 15.3 million patients fall into one of the following categories 1) They have de-registered with the practice contributing to IMRD 2) Their practice has stopped contributing to IMRD 3) They have requested to no longer contribute data to IMRD 4) They have died.

4. Table 1 - please give the full definition of 'IRR'

Response:

Incidence rate ratio. Please see line 144

5. Figure 1 and Figure 3 - can the authors explain the apparent peak in incidence in 2006?

Response:

This is an interesting observation and one we had not picked up on earlier. There does seem to be a one-off spike in incidence for adolescents in 2006 but not for other age groups (fig.1). However, the confidence intervals overlap with both 2005 and 2007. Additionally, given that we identified only 93 incident adolescent cases during 2003, and a similar spike was not seen in other age groups, we are unable to conclude that this is of any significance.

6. Can the authors offer some suggestion as to the varying geographical incidence? Why might incidence be rising more quickly in different regions for UC vs. CD? Or in the devolved nations vs. rest of the UK for CD?

Response:

We did not report on temporal trends in incidence stratified by geographical region in the present manuscript. However, we have expanded the discussion to include a section on geographical differences in overall incidence. [lines 270-274]

Reviewer: 3

1) Page 4, line 24 to 27: "The date of meeting ACU was the date after which the practice was confirmed to have an average of at least one medical record, one additional health record and two prescriptions per person-year."

I think something is missing from this sentence relating to acceptable computer usage - presumably this should relate to medical records, health records and/or prescriptions being computerised?

Response:

That is correct, we have now amended to clarify [line 85]

2) Page 4, line 31: "The study was a dynamic cohort with individuals entering and exiting at different times" and line 36-37: "Cohort exit was defined as the earliest date of the following: first diagnosis of IBD; de-registration with the GP practice contributing data...."

Could an individual enter and exit the cohort multiple times (e.g. with unique individuals identified by a single anonymised ID?) for example if an individual left one GP practice and registered at another?

Response:

Individuals could not leave then re-enter the cohort. If an individual de-registered with one practice contributing data and registered with another then they would be allocated a new unique identifier. Please see earlier response for reviewer 2 (response 1)

3) Outcome definitions according to the algorithm - presumably this algorithm did not diagnose IBD based on the prescriptions alone (i.e. point 2) and a Read code of symptoms was required (i.e. point 1)?

The paragraph below referring to a 'quality filter' suggests that first the algorithm validated the diagnosis (based on points 1 or 2 or both?) and then this quality filter checks what the algorithm has diagnosed? I don't think this is the approach taken, I think the definition here is actually what the 'quality filter' apparently does: i.e. "individuals were only included in the study as cases if they had at

least two IBD Read codes recorded on separate dates or at least one IBD Read code plus at least one prescription for a drug commonly used to treat IBD."

So to summarise, the descriptions of the algorithm and the quality filter are somewhat confusing. Could these paragraphs be clarified please?

Response:

Correct, individuals needed either two Read codes or one Read code plus one prescription code [lines 104-104]. Agreed, these paragraphs were confusing as there is overlap between the 'quality filter' and the 'algorithm'. The 'algorithm' was essentially a separate piece of work to explore the validity of the diagnosis of IBD in IMRD. Subjects do not need to satisfy the 'algorithm criteria' to qualify as an IBD case. [lines 117-118]

4) Statistical analysis, page 5, line 16-17: "Time period was fitted as both a continuous variable and a categorical variable by calendar year. Mixed multivariable Poisson regression was used to estimate incidence rate ratios (IRRs)."

Please clarify how time period was analysed as a continuous variable, I cannot see any results for this within the results section?

Within the mixed Poisson regression model, if GP practice was included as a random effect, were other covariates included as fixed-effects?

Response:

Time period was analysed as a continuous linear variable to report average change in incidence per calendar year. This was not included in the results table but throughout the body of the text. We have now added to the results table. Yes, the other covariates were included as fixed effects and we have updated the methods section [line 146]

5) Results: Table 1 and Figures 1 to 3 present incidence rates and IRRs of age groups and by calendar year yet the main findings of this study seem to relate to changes in incident rates over time and within age groups. Some numerical results are quotes for these findings in the text only and the key finding mentioned within the discussion and conclusion seems to be "we observed a 38% fall in incidence for those aged over 40 years and a 94% rise in incidence in the adolescent population" - yet I can't find the numerical results which relate to this finding in the results section?

I would like to see more detail of the age-time interactions described within the text added to Table 1 (or an additional table if there is not enough space), also including the numerical values of these interaction analyses for the age groups where the incident rates remain 'stable' for comparison and for completeness.

Response:

Thank you for highlighting. We have now included the numerical results to clarify these findings [lines 184 to 186].

We have added a table in the appendix with age-time interaction outputs from the model (lines 188-190)

6) Results: It may be helpful to include a flow diagram or similar to help with visualising the numbers at the start of the results relating to the total number in the cohort, the number of cases and the number of incident cases (of any IBD, of CD and of UC)

I also don't understand the difference between the 22,560 incident diagnoses mentioned in the first paragraph of the results and the 28,879 individuals given a new code for IBD. Please clarify the difference between these numbers and these definitions.

Response:

We think the flow diagram is a great idea, but with a figure limit of four, we have decided to prioritise graphs showing temporal trends in incidence and geographical distribution of disease. We have instead expanded the first paragraph of the results to make cohort size and case numbers as clear as possible [lines 166-169].

The 22,560 incident diagnoses are the number of individuals who met our criteria for incident IBD diagnoses during study follow up (One diagnostic Read code plus another on a subsequent visit or one Read code plus one prescription). The 28,879 are the individuals who had one Read code during study follow up.

7) Table 1: The following additions would be helpful for interpretation:

The number of cases with within the covariate subgroups (i.e. how many males and females? How many people in each age group etc.) A footnote to add the interpretation of the adjusted IRR - e.g. what does an $IRR < 1$ or > 1 correspond to?

Also, is there any particular reason that the age group 17-40 is the reference group? Is this part of the Montreal/Paris definition?

Response:

We are very happy to share these data with you. However, if we were to include the number of cases then we would also need to include the denominator size for the subgroups for comparison. We feel that the addition of three new columns in this large table would make it difficult to interpret. We have added a footnote to table 1 as requested.

The age group 17-40 years was selected as the reference group as this is simply the most common age to develop IBD in the Montreal/Paris classification system. Although the 0-9 years group comes first in the table, as IBD is very rare in the under 10s we felt it would not be intuitive to use this as the reference group.

8) UC incidence, page 8: Fig.1 is referred to in the first paragraph but I think this should be a reference to Figure 2?

Response:

Correct, we have amended [line 212]

Reviewer: 4

Chi-trend tests could have been used to analyse the trend of the incidence rates by year.

Response:

Although chi-trend tests may have helped to analyse trends over time, we used Poisson regression to analyse temporal trends in incidence and also presented crude incidence rates by calendar year. This is the approach taken in a number of similar studies [2, 3].

Reviewer: 5

1. Was the algorithm used to identify IBD patients validated? How sensitive and specific is the algorithm for identifying true IBD cases? The authors mention that "the diagnosis of IBD was validated

in a similar GP database" but no details are given. It would be helpful to know the possible extent of misclassification.

Response:

The details of the previous validation study are given in the 'Data source' section of the methods [line 79] [4]. Due to resource constraints, we were unable to validate our diagnostic algorithm by interrogating free-text notes. However, we anticipate that it is at least 92% specific; the previous validation study found diagnostic codes alone to give a 92% positive predictive value for the individual having IBD and our study had stricter diagnostic criteria.

2. I'm unclear as to how newly diagnosed cases were differentiated from existing cases. A person could have two IBD Read codes or an IBD Read code plus prescription but also have been diagnosed at a previous date (e.g., prior to 2000). Please clarify how cases were determined to be incident cases.

Response:

The individuals were only counted as incident cases if these were new diagnostic codes recorded during the study period and they had never received an IBD code before (we had access to all medical records so looked back prior to study start). We also only enrolled patients at an earliest date of nine months after GP registration to account for prevalent disease being recorded as incident disease when patients register with the practice. Nine months was chosen using Lewis plots to determine the appropriate time period [1]. Therefore, we are confident that our cases represent true incident cases and not prevalent disease.

3. Please clarify for readers who are less familiar with IBD: At what age does the disease typically develop? Is it more likely to develop during certain times in the lifespan? If so, could this explain some of the trends? For example, if most people develop the disease when they are young, perhaps the decreasing incidence rates in 40+ and increasing rates in 10-16 year-olds simply reflects improved diagnosis whereby true cases are being caught earlier. Please provide this background information for readers who are less familiar with IBD.

Response:

We agree. We have modified the discussion to include this information [line 292-299]

4. I found these two sentences confusing, so would suggest rewriting these for clarity: "The phenomenon of compounding prevalence of IBD has been demonstrated in Canada and in Scotland.[10, 32]. This relates to the principle that although IBD incidence may be static or falling, whilst IBD mortality remains very low, prevalence will increase; In Scotland IBD prevalence is estimated to reach 1.0% by 2028."

Response:

We have reworded accordingly [line 309-311]

Reviewer: 6

1. Please provide a sample description of the 11,325,025 individuals involved in the analyses (e.g. numbers of incident cases, prevalent cases and no diagnosis; numbers of males and females; age at study entry, age at diagnosis, etc.).

Response:

We have expanded the description of our cohort in the first paragraph of the results. Given that we observed a bimodal distribution for age at diagnosis, including median age at diagnosis may be misleading. A graph showing IBD incidence relative to age at onset of disease included in the appendix.

2. Please include the average (and median) follow up time among the prevalent IBD cases, incident IBD cases and those without the diagnosis in the result section.

Response:

We have added results on length of follow up for the whole cohort [line 165-169]. However, as individuals exited the cohort on diagnosis, we have not included length of follow up for the specified sub-groups.

3. To benefit the readers, please indicate how the crude incidence rates were estimated in the method section. Were follow up time of those prevalent cases removed from the denominator when calculating the incidence rates?

Response:

This has been added [lines 139-140]. Yes, prevalent cases were removed from the denominator for incidence calculations.

4. What was the denominator for the point prevalence calculation (i.e. no. of individuals contributing data to the cohort on the last day of the study period)? Were deceased cases of IBD excluded from the point prevalence calculation?

Response:

The denominator for the prevalence calculation was all living individuals contributing to the study on the last day of the study period [lines 148-149]. Deceased individuals were not included.

5. Were the average rates of increase/decrease per calendar year presented throughout the result section controlled for other covariates listed in the method section? If so, please make it clear.

Response:

We agree. This has now been changed [line 145]

VERSION 2 – REVIEW

REVIEWER	Takashi Ishige Gunma University Graduate School of Medicine, Japan
REVIEW RETURNED	21-May-2020

GENERAL COMMENTS	The revised manuscript clearly reflects the reviewers' suggestions. I c
---

REVIEWER	Jennie Clough Guy's and St Thomas' NHS Foundation Trust, UK
REVIEW RETURNED	21-May-2020

GENERAL COMMENTS	I am satisfied that the authors have addressed the comments in my initial review, and that the resubmission represents a manuscript suitable for publication.
---

REVIEWER	Sarah Nevitt University of Liverpool United Kingdom
REVIEW RETURNED	27-May-2020

GENERAL COMMENTS	Thank you to the authors for their responses and for their efforts in addressing my comments. I am satisfied that all comments have been sufficiently addressed and I am happy that the authors feel the comments have significantly improved their manuscript. I am happy to recommend this work for publication
---

REVIEWER	Kristin Sainani Stanford University, U.S.
REVIEW RETURNED	21-May-2020

GENERAL COMMENTS	The authors have adequately addressed my concerns.
--

REVIEWER	Yue-Fang Chang University of Pittsburgh, USA
REVIEW RETURNED	04-Jun-2020

GENERAL COMMENTS	Comments have been addressed satisfactorily.
--